# Validation of asthma recording in electronic health records: protocol for a systematic review

Francis Nissen,[1] Jennifer K Quint,[2] Samantha Wilkinson,[1] Hana Mullerova,[3] Liam Smeeth,[1] Ian J Douglas[1]

► Prepublication history and additional material are available. To view these files please visit the journal online (http://dx.doi.org/10.1136/bmjopen-2016-014694).

[1]Department of Non-Communicable Disease Epidemiology, London School of Hygiene and Tropical Medicine, London, UK
[2]National Heart and Lung Institute, Imperial College, London, UK
[3]RWD & Epidemiology, GSK R&D, Uxbridge, UK

**Correspondence to**
Dr Francis Nissen;
francis.nissen@lshtm.ac.uk

## ABSTRACT

**Background** Asthma is a common, heterogeneous disease with significant morbidity and mortality worldwide. It can be difficult to define in epidemiological studies using electronic health records as the diagnosis is based on non-specific respiratory symptoms and spirometry, neither of which are routinely registered. Electronic health records can nonetheless be valuable to study the epidemiology, management, healthcare use and control of asthma. For health databases to be useful sources of information, asthma diagnoses should ideally be validated. The primary objectives are to provide an overview of the methods used to validate asthma diagnoses in electronic health records and summarise the results of the validation studies.

**Methods** EMBASE and MEDLINE will be systematically searched for appropriate search terms. The searches will cover all studies in these databases up to October 2016 with no start date and will yield studies that have validated algorithms or codes for the diagnosis of asthma in electronic health records. At least one test validation measure (sensitivity, specificity, positive predictive value, negative predictive value or other) is necessary for inclusion. In addition, we require the validated algorithms to be compared with an external golden standard, such as a manual review, a questionnaire or an independent second database. We will summarise key data including author, year of publication, country, time period, date, data source, population, case characteristics, clinical events, algorithms, gold standard and validation statistics in a uniform table.

**Ethics and dissemination** This study is a synthesis of previously published studies and, therefore, no ethical approval is required. The results will be submitted to a peer-reviewed journal for publication. Results from this systematic review can be used to study outcome research on asthma and can be used to identify case definitions for asthma.

**PROSPERO registration number** CRD42016041798.

### Strengths and limitations of this study

► To our knowledge, this is the first systematic review to identify and evaluate methods used to validate a recording of asthma diagnosis in electronic health records.
► The review of validation of asthma diagnosis in electronic health records could inform selection of asthma identification algorithms used by future health outcome studies and identify any gaps in quality and scope of validation studies. It will also provide an overview of the algorithms with their positive predictive value, negative predictive value, sensitivity or specificity.
► Different databases may validate different algorithms to identify asthma, which might limit the generalisability of these algorithms as they are context-specific.
► This review is focused on the methodology of asthma recording validation, and not on all outcome results of studies (except the validation results). Because of this, publication bias might be an issue (methods that do not find positive results may be less likely to have been published).

## BACKGROUND

Asthma is a common chronic inflammatory disease of the airways. This condition is characterised by a variable expiratory airflow limitation which is generally reversible. The core symptoms are cough, wheeze, breathlessness and chest tightness.[1] Asthma episodes can range from mild attacks, which interrupt daily life and work productivity, to severe and life-threatening attacks.[2] Asthma is inherently variable and individuals will experience fluctuating symptoms. Most commonly, asthma emerges in childhood, but it can also arise in adulthood. Therefore, adult asthma consists of both persistent or relapsed childhood disease and true incident adult disease. There is no cure, but with the right treatment, symptoms can usually be managed and patients with asthma can lead their lives without disruption.[1]

The widespread adoption of electronic health records (EHRs) means that large population-based primary and secondary care databases are available, proving a great opportunity for research on asthma and other diseases. The availability of routinely generated longitudinal records for research has dramatically increased over the last

decades.[3] However, the primary function of EHRs is to support healthcare clinical decision-making, not research purposes. The integrity of the research generated from EHRs may be questionable, unless data are thoroughly validated for this purpose.[4–7]

EHRs are a digital reflection of the paper medical chart, while the main purpose of administrative claims data is administration of reimbursements to healthcare providers for their services. This systematic review will only consider data from EHR as the quality measures between the two types of data can be markedly different.[8 9]

EHRs store information about diagnoses as clinical codes. A single code, or an algorithm consisting of multiple codes, can be used to retrieve records from EHR, and additional restrictions can be applied such as age or exclusion of other diseases.[7 10] Alternatively, several authors have recently used natural language processing and machine learning techniques to automate algorithm generation for the identification of asthma diagnoses from large databases.[11–13] The most common method to

assess the validity of algorithms is to compare them with a gold standard such as another linkable data set or request a verification from the treating physician or the patient via a questionnaire.[10] Another approach is active case detection where the databases are constantly screened to identify cases that emerge.[14]

Several limitations apply to the validation of diagnosis recording in EHR. First, individual databases often only cover a single-care setting (primary or secondary care) as such case ascertainment only relies on a partial description of the healthcare pathway.[15] Another issue is that the validity of different diseases will not necessarily be the same in a given data set. For example, mental health disorders such as anxiety or depression might be coded using less specific symptoms, whereas the validity of diagnoses with a very high specificity such as breast cancer is likely to be superior. There have been multiple studies which have measured the validity of specific databases for asthma.[16 17] Sharifi *et al* have conducted a systematic review on validated methods to

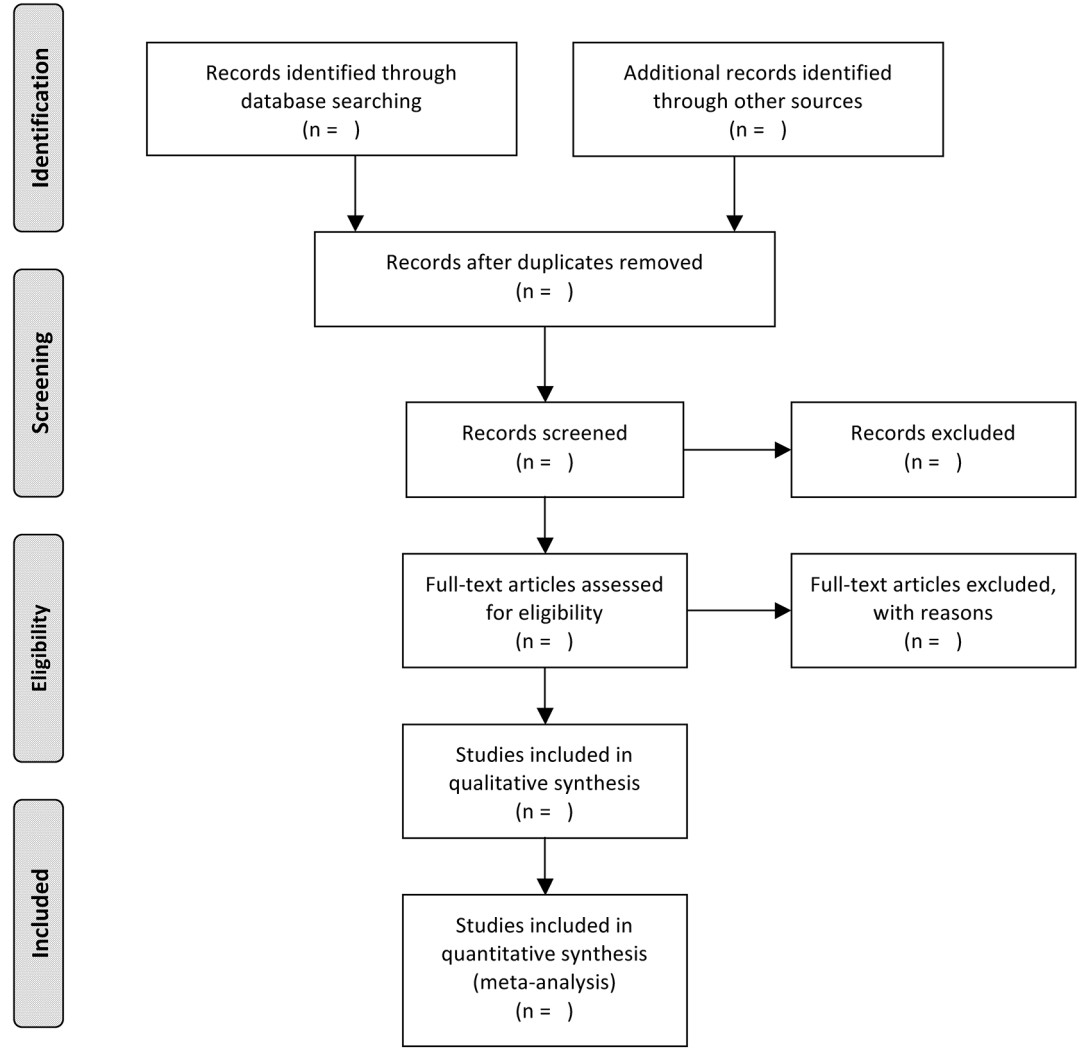

**Figure 1** Study screening process: Preferred Reporting Items for Systematic Reviews and Meta-Analyses flow diagram from Moher *et al*.

capture acute bronchospasm using administrative or claims data,[18] which yielded two validation studies of bronchospasm codes.[11 19]

This systematic literature review aims to provide an overview of methods used to validate asthma diagnoses, specifically in EHR. Such a study has not yet been published in the medical literature to the best of our knowledge.

## Research question

The primary objectives of this systematic review are to provide an overview of both the methods with which asthma diagnosis recording has been validated in EHR and the estimates of the validation test measures.

The questions of interest for this systematic review are

1. Which EHRs that are not only based on claims data have been used to obtain information on the diagnosis of asthma?
2. Which algorithms have been used to define an asthma diagnosis (including diagnostic codes, possible spirometry tests and clinical descriptions)?
3. How were the diagnostic criteria applied to the data sources and which other approaches have been used to validate a case definition?
4. What are the estimates for the positive predictive value, negative predictive value, specificity and sensitivity for a diagnosis of asthma in EHRs that are not solely claims-based?

## METHODS

MEDLINE and EMBASE will be searched for the terms 'asthma', 'validation', 'electronic databases' and synonyms for each of these terms. In addition, reference lists of review articles and retrieved articles will be reviewed. The Preferred Reporting Items for Systematic Reviews and Meta-Analyses flow diagram of this protocol, from Moher et al,[20] can be found in figure 1, and the search strategy can be found in the online supplementary file.

## Inclusion criteria

Any type of observational study design that used EHR to validate the recording of an asthma diagnosis will be considered. Articles will only be considered if published in English and before October 2016 without any specific start date. Within the databases, we will consider asthma diagnoses based on both structured data (such as laboratory results and prescriptions) and free text data (such as spirometry results). We require the validated algorithms to be compared with an external gold standard, such as a manual review, questionnaires (completed by the patient or their physician) or an independent second database. We will include algorithms formed of single codes, those requiring multiple case characteristics and algorithms generated by natural language processing or machine-learning.

## Exclusion criteria

Studies which involve pharmacovigilance databases (signal detection or spontaneous reporting), studies without validation process of asthma recording and conference abstracts will be excluded. Algorithms used in databases originating from only claims data will also be excluded as a systematic review on the validated methods to capture acute bronchospasm using claims data has been published recently.[18]

Two independent authors will scan the abstracts and titles against the research questions and exclusion criteria and select articles for full-text review. After this full-text article review, eligibility for inclusion in the report will be decided by consensus or arbitration by a third reviewer. A uniform table with information of each included study will be populated after data extraction, which will include information on the author, date of publication, journal, database, algorithms, population, gold standard and test measure(s).

## Data synthesis

Studies and study data will be managed using EndNote and Microsoft Excel, respectively. The methods for asthma recording validation will be summarised in a narrative synthesis and tables describing all identified verification processes, and their results. These results will consist of the recorded PPV, NPV, sensitivity and specificity of the included studies. Where possible, these tests will be calculated if they are not reported within the study.

**Contributors** JQ, ID, LS and HM were responsible for developing the research question and have advised on the data collection and search strategies. FN drafted the manuscript. FN and SW reviewed the literature and summarise the found papers. ID is responsible for study management and coordination. All authors have read, commented on and approved the final manuscript.

**Funding** This work was supported by GlaxoSmithKline (GSK), through a PhD scholarship for FN with grant number EPNCZF5310. The publishing of this study was supported by the Wellcome Trust: grant number 098504/Z/12/Z.

**Competing interests** FN and SW are funded by a GSK scholarship during their PhD programmes. JQ reports grants from MRC, BLF, Wellcome Trust and has received research funds from GSK, AZ, Quintiles IMS and had personal fees from AZ, Chiesi, BI. HM is an employee of GSK R&D and owns shares of GSK Plc. ID is funded by, holds stock in and has consulted for GSK.

**Provenance and peer review** Not commissioned; externally peer reviewed.

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
