## [Reviewer comments · BMJ Open]

ARTICLE DETAILS

TITLE (PROVISIONAL)	Validation of asthma recording in electronic health records: protocol for a systematic review
AUTHORS	Nissen, Francis; Quint, Jennifer; Wilkinson, Samantha; Müllerová, Hana; Smeeth, Liam; Douglas, Ian

VERSION 1 - REVIEW

REVIEWER	Liisa Jaakkimainen Institute for Clinical Evaluative Sciences, Toronto, Canada Department of Family and Community Medicine, University of Toronto, Toronto, Canada
REVIEW RETURNED	29-Nov-2016

GENERAL COMMENTS	This is a well written and important proposal of a systematic review of EHR algorithms to identify people with asthma. A few minor suggestions below: 1. I would clarify the terms EHR vs EMR. Is proposed search going to include clinical EMR data (ie physician notes) or health administrative data (claims, hospital admission data) or both?2. EHRs contains both unstructured (results of spirometry tests) and structured data (lab results, prescriptions), as well as coded data (billing or claims codes). While there is a mention of examining the type of databases to be included in the review, I would clarify what types of data within the databases will be examined in the systematic review.3. The objective to validate methods used to validate asthma diagnoses seems appropriate. But in other areas of the document (ie the strengths section), it seems that this is a validation of asthma codes? I think the goal would be to do a review of algorithms to identify people with asthma and how well asthma is coded in the EHR? Maybe some consistency in proposal would clarify which is the objective.4. Not sure what "non-population" based studies would be excluded? Would these be from speciality clinics? Maybe an example here would be helpful.5. The quality of the EHR studies are likely going to be different from judging the quality of studies only using health administrative data. It's just not quite clear if there is a study quality measure for EHR studies that will be used.
--

REVIEWER	Hongfang Liu Mayo Clinic, USA
REVIEW RETURNED	07-Dec-2016

GENERAL COMMENTS	This manuscript describes the “protocol” for a systematic review developed by authors - actual systematic review has not been performed and reported yet. This systematic reviews aim to summarize methods (definitions) and test measures of asthma diagnosis validation in EHRs. Authors claim that this is the first systematic review for these purposes. Since it has been reported that asthma has inconsistent ascertainment process and criteria causing delayed diagnosis, this systematic review would be useful in the community. However, there are some major concerns about this manuscript as follows: Major: This manuscript describes the high-level “protocol” without any results, not even the result of the initial queries against the data sources. The proposed protocol has not been demonstrated at all through any outcomes and so it is hard to know if this systematic review would be well feasible and produce reasonable outcomes. One inclusion criterion is the use of reference standard case definition for asthma diagnosis – i.e., the definition of asthma by Global Initiative for Asthma, but this definition is not been described in the manuscript and so hard to judge whether its use is appropriate or not. Asthma diagnosis has known to be inconsistent. Using only this definition may exclude other asthma diagnoses and bias the outcomes of systematic review. The inclusion/exclusion criteria are not explained details to be replicated. For example, reference standard case definition for asthma diagnosis, how to find non-population-based studies. As the recent growth of EHRs, utilizing natural language processing and machine learning techniques to automate asthma diagnosis have been studied. How do authors consider these studies in their systematic reviews along with traditional manual chart review and code-based approaches? Minor: Use abbreviation once it’s defined such as EHR.
---

REVIEWER	Mome Mukherjee Asthma UK Centre for Applied Research (AUKCAR), Centre for Medical Informatics, Usher Institute of Population Health Sciences and Informatics, Medical School, University of Edinburgh, Edinburgh, EH8 9AG, UK
REVIEW RETURNED	12-Dec-2016

GENERAL COMMENTS	This study will be a valuable piece of work. Some minor comments - i) PROSPERO mentions studies with participants age 18 and over, but the protocol does not mention any age criterion. If there is any deviation, it needs mentioned ii) the appendix mentioned contains search terms for the review and the STARD criteria as mentioned iii) reference is made to the earlier STARD initiative but not to STARD 2015.
---

	The reviewer also provided a marked copy with additional comments. Please contact the publisher for full details.
--	---

VERSION 1 – AUTHOR RESPONSE

Reviewer: 1

Reviewer Name: Liisa Jaakkimainen

Institution and Country: Institute for Clinical Evaluative Sciences, Toronto, Canada, Department of Family and Community Medicine, University of Toronto, Toronto, Canada

This is a well written and important proposal of a systematic review of EHR algorithms to identify people with asthma.

A few minor suggestions below:

1. I would clarify the terms EHR vs EMR. Is proposed search going to include clinical EMR data (ie physician notes) or health administrative data (claims, hospital admission data) or both?

--We have added a paragraph on the issue of EHR vs administrative data on page 5. Only studies which draw their data from EHR will be reviewed.

2. EHRs contains both unstructured (results of spirometry tests) and structured data (lab results, prescriptions), as well as coded data (billing or claims codes). While there is a mention of examining the type of databases to be include in the review, I would clarify what types of data within the databases will be examined in the systematic review.

--We have specified the type of data used in the inclusion criteria on page 8.

3. The objective to validate methods used to validate asthma diagnoses seems appropriate. But in other areas of the document (ie the strengths section), it seems that this is a validation of asthma codes? I think the goal would be to do a review of algorithms to identify people with asthma and how well asthma is coded in the EHR? Maybe some consistency in proposal would clarify which is the objective.

--We have made the objectives more consistent throughout the document.

4. Not sure what “non-population” based studies would be excluded? Would these be from speciality clinics? Maybe an example here would be helpful.

--We have excluded the non-population-based requirement. Some studies could draw their study population from people for whom they have samples available (such as DNA samples), or based on the inclusion criteria of previous studies whose asthma diagnoses would now be validated. To make the systematic review more comprehensive, we have removed the non-population based criterion.

5. The quality of the EHR studies are likely going to be different from judging the quality of studies only using health administrative data. It’s just not quite clear if there is a study quality measure for EHR studies that will be used.

--We have added a paragraph on EHR vs administrative data on page 5, only studies using EHR data will be reviewed..

Reviewer: 2

Reviewer Name: Hongfang Liu

Institution and Country: Mayo Clinic, USA

This manuscript describes the “protocol” for a systematic review developed by authors - actual systematic review has not been performed and reported yet. This systematic reviews aim to summarize methods (definitions) and test measures of asthma diagnosis validation in EHRs. Authors claim that this is the first systematic review for these purposes. Since it has been reported that asthma has inconsistent ascertainment process and criteria causing delayed diagnosis, this systematic review would be useful in the community. However, there are some major concerns about this manuscript as follows:

Major:

This manuscript describes the high-level “protocol” without any results, not even the result of the initial queries against the data sources. The proposed protocol has not been demonstrated at all through any outcomes and so it is hard to know if this systematic review would be well feasible and produce reasonable outcomes.

--Using the search strategy outlined in the protocol, we found 332 citations in Medline and 1014 in Embase. Of those, 946 were non-duplicate among the two databases. These feasibility counts are not always published in the protocol itself.

One inclusion criterion is the use of reference standard case definition for asthma diagnosis – i.e., the definition of asthma by Global Initiative for Asthma, but this definition is not been described in the manuscript and so hard to judge whether its use is appropriate or not. Asthma diagnosis has known to be inconsistent. Using only this definition may exclude other asthma diagnoses and bias the outcomes of systematic review.

--We agree with this point and we have now removed the requirement for a standard case definition.

The inclusion/exclusion criteria are not explained details to be replicated. For example, reference standard case definition for asthma diagnosis, how to find non-population-based studies.

--We have removed both requirements, as outlined above.

As the recent growth of EHRs, utilizing natural language processing and machine learning techniques to automate asthma diagnosis have been studied. How do authors consider these studies in their systematic reviews along with traditional manual chart review and code-based approaches?

--We would include natural language and machine learning processes if they present. We have specified this in both the background and methods section in the amended version.

Minor:

Use abbreviation once it's defined such as EHR.

--This has been addressed.

Reviewer: 3

Reviewer Name: Mome Mukherjee

Institution and Country: Asthma UK Centre for Applied Research (AUKCAR), Centre for Medical Informatics, Usher Institute of Population Health Sciences and Informatics, Medical School, University of Edinburgh, Edinburgh, EH8 9AG, UK

This study will be a valuable piece of work.

Some minor comments - i) PROSPERO mentions studies with participants age 18 and over, but the protocol does not mention any age criterion. If there is any deviation, it needs mentioned

--We have updated the PROSPERO submission, this systematic review was registered on PROSPERO in a preliminary stage.

ii) the appendix mentioned contains search terms for the review and the STARD criteria as mentioned

--We have removed the STARD requirement in the amended version, as it is only for quality of reporting in validation studies in claims data. The checklist would be impractical as it was intended for

administrative databases with a strict standard case definition, which we have removed in the amended version.

iii) reference is made to the earlier STARD initiative but not to STARD 2015.

--We have removed the STARD requirement, as mentioned above.

Editor Comments to Author:

At the moment, you use STARD to assess quality of studies, but this is not correct, as this is for the quality of reporting. Please use an alternative tool.

--We have removed the STARD requirement, as outlined above. In addition to the raised point that it only addressed the quality of reporting, the checklist would be impractical as it was intended for use with administrative data and with a standard case definition, which we have removed from the requirements.

Please give the start date for the searches - is this all time? The PROSPERO entry gives a date, but this is not in the paper itself.

--We have amended the study to reflect that there is no start date.

Please include and fill out a PRISMA-P checklist with page numbers.

--We have included the PRISMA-P checklist as a supplementary file.

The aim is repeated twice in abstract - delete one.

--We have addressed this issue.

The start date is needed in the abstract.

--We have amended the abstract to reflect that there is no start date.

VERSION 2 – REVIEW

REVIEWER	Liisa Jaakkimainen Institute for Clinical Evaluative Sciences Toronto, Canada
REVIEW RETURNED	14-Feb-2017

GENERAL COMMENTS	I feel the authors have addressed the questions I had in the first review.
--

REVIEWER	Mome Mukherjee University of Edinburgh, UK
REVIEW RETURNED	09-Feb-2017

GENERAL COMMENTS	Few minor comments: 1) Although distinction has been made between claims-EHR and EHR for paper medical chart, in the first research question and in review aims, it will help for clarification purpose if "Which EHR..." and "...EHR" respectively, refer to non-claim EHR. Since EHRs mean different things in different places (Chapter 6 of https://tinyurl.com/zxbrhjr), I would strongly recommend putting in a definition of EHR that the researchers are using. 2) Should 'diagnosis' not be used as a search term, when that is the purpose of the review? If not, providing a justification would be helpful. 3) Spirometry results have been provided as an example for "unstructured data". But spirometry results could be numerical and/or free text. I think the author/s meant free text data for "unstructured data".
--

VERSION 2 – AUTHOR RESPONSE

Thank you for your review, please find our response below.

- 1) This is a valid point. It is not clear from the research question alone that we are only reviewing validation studies that are not solely based on claims data. The research question has been amended to reflect this.
- 2) We would prefer to keep the systematic review as comprehensive as possible. In addition, we also want to review studies that have validated asthma recording, even if the main aim of the study was different. These studies might not specifically state "asthma diagnosis". Studies that do not validate an asthma diagnosis can still be manually excluded.
- 3) We agree, "unstructured" has been changed to "free text" in the amended document.

VERSION 3 – REVIEW

REVIEWER	Liisa Jaakkimainen Department of Family and Community Medicine University of Toronto Institute for Clinical Evaluative Sciences Toronto, Ontario Canada
REVIEW RETURNED	20-Apr-2017

GENERAL COMMENTS	I feel the authors have improved the paper and addresses reviewers concerns.
--